# Residue and Risk Assessment of Fluopyram in Carrot Tissues

**DOI:** 10.3390/molecules27175544

**Published:** 2022-08-29

**Authors:** Yiyue Yang, Ming Yang, Tong Zhao, Lingyi Pan, Li Jia, Lufei Zheng

**Affiliations:** 1Institute of Quality Standard and Testing Technology for Agro-Products, Chinese Academy of Agricultural Sciences, Beijing 100081, China; 2Institute of Apicultural Research, Chinese Academy of Agricultural Sciences, Beijing 100093, China

**Keywords:** fluopyram, carrot, LC-QQQ-MS/MS, pesticide residue analysis, pesticide risk assessment

## Abstract

This study describes the variation in residue behavior of fluopyram in soil, carrot root, and carrot leaf samples after the application of fluopyram (41.7% suspension, Bayer) by foliar spray or root irrigation at the standard of 250.00 g active ingredient per hectare (a.i./ha) and double-dose treatment (500.00 g a.i./ha). Fluopyram and its metabolite fluopyram-benzamide were extracted and cleaned up using the QuEChERS method and subsequently quantified with LC-QQQ-MS/MS. The LOD and LOQ of the developed method were in the range of 0.05–2.65 ug/kg and 0.16–8.82 ug/kg, respectively. After root irrigation, the final residues detected in edible parts were 0.60 and 1.80 mg/kg, respectively, when 250.00 and 500.00 g a.i./ha were applied, which is much higher than the maximum residue limit in China (0.40 mg/kg). In contrast, after spray application, most of the fluopyram dissipated from the surface of carrot leaves, and the final residues in carrot roots were both only 0.05 mg/kg. Dietary risk assessments revealed a 23–40% risk quotient for the root irrigation method, which was higher than that for the foliar spray method (8–14%). This is the first report comparing the residue behavior of fluopyram applied by root irrigation and foliar spray. This study demonstrates the difference in risk associated with the two application methods and can serve as a reference for the safe application of fluopyram.

## 1. Introduction

Carrot (*Daucus carota* L.) is one of the most important root vegetables in the Apiaceae family; it is cultivated and consumed worldwide [1]. The health benefits of carrot roots are attributed to its abundance of provitamin A, carotenoids, and dietary fiber, as well as the numerous minerals and antioxidants it contains [2,3]. According to the global production records of primary vegetables, carrots are among the top five vegetables [4], and the import and export quantity of carrots has risen steadily over the past 40 years, with a commensurate 11-fold increase in its value. Thus, the carrot is of great economic and agricultural importance.

The quality of carrot crops can be affected by pests and fungal pathogens, such as the carrot fly (*Psila rosae*), the soilborne fungus *Sclerotinia sclerotiorum*, which causes black root rot and crown rot, and *Pythium violae*, which causes cavity spots [5]. Pesticides are often applied to carrot crops to control these diseases and reduce loss [5,6]. Recently, a 41.7% suspension concentrate (SC) of fluopyram, which has an effective control against many diseases, such as powdery mildew and root-knot nematode, has been registered in China. Its active ingredient, fluopyram (Figure 1a) (N-[2-[3-chloro-5-(trifluoromethyl)-2-pyridinyl] ethyl]-2-(trifluoromethyl) benzamide), is a systemic broad-spectrum fungicide with preventive and curative properties [7]. The major metabolite, fluopyram-benzamide (Figure 1b), is used for risk assessment [8], and the sum of fluopyram and fluopyram-benzamide is expressed as fluopyram. Since 2013, fluopyram has been sold in more than 60 countries and regions, and it is registered for use in over 70 crops [9]. In China, the registration of fluopyram covers 26 crops, including carrot, cucumber, tomato, and apple [10].

Two application methods, foliar spray and root irrigation, are allowed for the application of fluopyram on crops in China [10]. Several studies have evaluated the residue behavior of fluopyram on fruits and vegetables after treatment by foliar spray or root irrigation [7,11,12,13,14,15,16]. For example, Matadha et al. studied the distribution of fluopyram on pomegranate tissue by spray application [7], and Chawla et al. investigated the residual behavior of fluopyram applied by root irrigation to cucumbers [12]. However, to the best of our knowledge, there is no report comparing the effects of the two application methods on the residual behavior and distribution of fluopyram in crops, and a single application method cannot fully reflect the mode of action and residual behavior of fluopyram on crops. Therefore, considering the mode of action of fungicides and that carrot is a root vegetable, it is meaningful and necessary to understand the residual behavior of fluopyram in carrots using the two application methods. This information could be used as a reference for the safe application of fluopyram to carrots and other vegetables.

Consequently, in this study, we conducted spray and root irrigation field trials of fluopyram on carrots according to the “Guideline on Pesticide Residue Testing” of China, and the residues of fluopyram in different carrot tissues and soil over time were measured using QuEChERS in conjunction with liquid chromatography-triple quadrupole tandem mass spectrometry (LC-QQQ-MS/MS) analysis. The data were useful for elucidating residue behavior and distribution of fluopyram in carrot roots, carrot leaves, and soil under the two application methods, as well as for conducting a risk assessment of the two application methods.

## 2. Results and Discussion

### 2.1. Results of Method Validation

The methods used in this study for the analysis of fluopyram and fluopyram benzamide in carrot roots, carrot leaves, and soils were evaluated with the following validation parameters: recovery, LOD, LOQ, linearity and precision, and satisfactory results were obtained for all the parameters studied (Table 1). When spiked at 0.01–1 mg/kg, the recoveries were 83–106% for fluopyram and 80–116% for fluopyram benzamide, and the LOD and LOQ of the developed method were in the range of 0.05–2.65 ug/kg and 0.16–8.82 ug/kg, respectively. The accuracy of the method was investigated by spiking the two analytes in carrot and soil, and satisfactory results were obtained. The method had good linearity when analyzing matrix-matched standards for the two analytes (R^2^ = 0.999).

### 2.2. Residues in the Soil

The residual levels of fluopyram in field soils after root irrigation and foliar spray at doses of 250.00 g a.i./ha and 500.00 g a.i./ha were initially 5.34 mg/kg and 14.46 mg/kg and 0.18 mg/kg and 0.33 mg/kg, respectively (Appendix A Appendix A). The large difference in initial residues may be caused by the differences in the application methods. Chawla et al. [12] have shown that root irrigation, in which pesticides are directly applied to the soil, results in high pesticide residues in the soil. During foliar spraying, most of the fluopyram remained on the surface of carrot leaves, and only a small amount of fluopyram was deposited in the soil [17]. It was observed that under the two application methods, the fungicides did not readily dissipate from the field soils—the final residue amounts were 6.02 mg/kg and 13.83 mg/kg for root irrigation and 0.30 mg/kg and 0.60 mg/kg for foliar spraying, respectively. These field studies were conducted in Beijing from July to August, which is the rainy season. The water solubility of fluopyram is only 16 mg/L and the vapor pressure is only 3.1 × 10^−6^ Pa at 25 °C [18]; therefore, fluopyram is less likely to volatilize from moist soils [19], resulting in consistently high residual levels in soils.

### 2.3. Residues in the Edible Part (Carrot Root)

Residual levels of fluopyram in carrot roots were determined after applications with the two methods at the standard (250.00 g a.i./ha) and double-dose treatments (500.00 g a.i./ha), starting at 10 min after application. The residual amount of fluopyram over time in carrot roots is shown in Figure 2. Initially, at 250.00 g a.i./ha, the average residues of fluopyram in carrot roots after irrigation and spray applications were 0.26 mg/kg and 0.13 mg/kg, respectively; at 500.00 g a.i./ha, the initial average residues were 0.56 mg/kg and 0.213 mg/kg, respectively (Table 2). The residues of fluopyram in carrot roots gradually increased over time after root irrigation, and the final residues were 0.60 mg/kg and 1.80 mg/kg, respectively (Table 2). In contrast, the residues of fluopyram in carrot roots after spray treatment began to decrease on day 3, and the final residues were both 0.05 mg/kg, respectively.

After root irrigation, the continuous increase in fluopyram concentration in the carrot roots may be related to a high amount of fluopyram residues in the soil. Crops typically absorb and accumulate pesticides from soil [16,20], and Lichtenstein et al. [21] reported that pesticide residues in carrots increase with increasing pesticide concentrations in soil. Thus, root crops such as carrots are more likely to absorb pesticides from the soil than above-ground vegetables [21,22]. In terms of pesticide properties, fluopyram is a lipophilic pesticide with an octanol–water partition coefficient (log Pow) of 3.3 (pH = 6.5), making it more likely to accumulate in the rhizomes of crops [18,23]. Vargas-Perez et al. [13] also reported that fluopyram is a persistent compound. Therefore, the authors concluded that the lipophilic pesticide fluopyram is more likely to accumulate in root crops, such as carrots, after root irrigation.

### 2.4. Residues in the Carrot Leaf

The initial residual concentration of fluopyram in carrot leaves was higher than that in the edible part of carrot (carrot root) under both application methods. Under the standard dose and double-dose treatments, the initial residues of fluopyram after root irrigation in the carrot leaves were 1.64 mg/kg and 3.37 mg/kg; the initial residues after spray application were 12.62 mg/kg and 23.55 mg/kg (Table 3). The residues dissipated much faster from carrot leaves than carrot roots under both application methods. Within only 7 days of application, 77–93% of the fluopyram had dissipated from leaves. On day 28, fluopyram residues in leaves were 0.13 and 0.53 mg/kg after root irrigation and 1.10 mg/kg and 1.44 mg/kg after spray application with the standard and double-dose treatments, respectively. This may be due to the fact that exposed carrot leaves receive light, unlike the deeply buried roots, and the photodegradation of pesticides on plant surfaces, especially leaves, is a major route for pesticide dissipation [24].

### 2.5. Half-Life (DT_50_)

Dynamic models of fluopyram in carrot samples following the two application methods are shown in Table 4. The dissipation of fluopyram from carrot roots after root irrigation did not follow first-order kinetics. After root irrigation at 250.00 g a.i./ha and 500.00 g a.i./ha, the half-lives of fluopyram in leaves were 9.1 days and 14.4 days, respectively, while the corresponding half-lives after spray application were 6.4 days and 5.5 days. The half-lives of fluopyram in spray-treated carrot roots were 14.1 days and 11 days, respectively, while the half-lives in irrigation-treated roots in this study were similar to cucumbers from the same treatment in another study [12]. According to another report on a foliar spray of fluopyram, the half-lives of fluopyram after the standard and double-dose treatments in onions were 8.9 days and 9.1 days [19]. Matadha et al. [7] reported that single and double-dose fluopyram half-lives in pomegranate fruit were 7.6 days and 8.6 days, while the half-life of fluopyram in watermelon was 6.48 days [25]. The half-life of fluopyram is affected by factors such as application dose, crop variety, and environmental factors (light, temperature, pH value, and humidity), and the half-life of fluopyram in carrot roots under the same treatment conditions was higher than that of the above crops [26].

### 2.6. Dietary Intake Risk Assessment of Fluopyram in Carrot

According to the food consumption habits of Chinese residents of different ages and sex and the fluopyram residues detected in this study, dietary intake risk assessments of parent fluopyram in carrots from the two application methods were carried out. Table 5 shows the results of the dietary intake risk assessment. Although the consumption of vegetables increased with age and body weight, the dietary intake risk gradually decreased. The dietary intake risk for children (2–12 years old) was significantly higher than any other age group, and the dietary risk was higher for carrots treated by root irrigation than foliar spray. Luckily, all calculated risk quotients ranged from 8% to 40%, which is below 100%. Therefore, these results indicate that fluopyram applied to carrots at the recommended application rate by root irrigation poses a higher threat to consumers through dietary intake than spray application, but both are within the acceptable range.

## 3. Materials and Methods

### 3.1. Chemicals and Reagents

Fluopyram (purity > 99%) and fluopyram benzamide (purity > 98.7%) were both obtained from Dr. Ehrenstorfer GmbH, Germany. A 41.7% fluopyram SC was procured from Bayer Crop Science, Beijing, China. Ammonium formate and formic acid were acquired from Thermo Fisher Scientific, Shanghai, China. Acetonitrile and methanol (chromatographic grade) were obtained from Mreda Technology Co., Ltd., Beijing, China. Anhydrous magnesium sulfate and sodium acetate were acquired from Sinopharm Chemical Reagent Co., Ltd., Beijing, China. Nylon membrane filters measuring 0.22 µm were procured from Agilent Technologies Inc., Santa Clara, CA, USA.

### 3.2. Preparation of Standard Solutions

Fluopyram and fluopyram benzamide were dissolved in acetonitrile to prepare 1000 mg/L standard stock solutions in 10 mL volumetric flasks and then diluted with carrot and soil blank sample extracts (i.e., matrix) to make 0.00500 mg/L, 0.0100 mg/L, 0.0500 mg/L, 0.1000 mg/L, 0.5000 mg/L, and 1.00 mg/L matrix-matching working standard solutions. The stock standard solutions and working standard solutions were stored in the dark at −20 °C.

### 3.3. Field Trial

Field trials were carried out in the Haidian District of Beijing from July to September 2021. In a 30 m^2^ plot area, a 41.7% fluopyram SC was applied to carrot fields by root irrigation and spray application. Pesticides were applied at the recommended dose of 250.00 g a.i./ha and at a double dose of 500.00 g a.i./ha when the carrot roots grew to the fleshy swelling stage. Carrots without the application of pesticides served as controls. The control carrots were separated from the treated carrots by a buffer zone. One kg of normal-growing carrots and soils were collected at 10 min, 2 h, 6 h, 1 day, 2 days, 3 days, 5 days, 7 days, 9 days, 11 days, 15 days, 21 days, and 28 days after application, using random sampling methods. After the sample was collected, the soil attached to carrot roots was removed with a dry rag or soft-bristled brush, and then the leaves and roots were separated and frozen in liquid nitrogen. Carrots and soil were stored at −20 °C until testing.

### 3.4. Sample Preparation

The carrots samples were chopped into small pieces and roots were homogenized in a homogenizer (Bear Electric Appliance Co., Ltd., Guangdong, China), while leaf samples were homogenized with a TARGIN multi-tube vortexer (Beijing Targin Technology Co., Ltd., Beijing, China). Soil samples from the field were air-dried in the laboratory at room temperature, powdered, and passed through a 2 mm sieve. The extraction of fluopyram from carrot tissues and field soil was conducted as previously described [13,19]. To 10.00 g carrot roots, carrot leaves and homogeneous soil samples 10 mL 1% acetic acid in acetonitrile were added in triplicate. Next, the samples were vortexed for 2 min, and then 6.00 g anhydrous magnesium sulfate and 1.50 g sodium acetate were added and dissolved by vortex mixing for 2 min. Samples were then centrifuged at 5000 r/min for 5 min, and the supernatant was filtered with a 0.22 μm nylon membrane and transferred to a vial for analysis.

### 3.5. Analysis of Fluopyram and Fluopyram Benzamide

The analysis of fluopyram and fluopyram-benzamide from carrot tissues and field soils after spray and root applications was carried out with an Agilent 6495 Liquid Chromatograph Triple Quadrupole Tandem Mass Spectrometer (LC-QQQ-MS/MS) (Agilent Corporation, Santa Clara, CA, USA). Separation was carried out on a HALO 160A, ES-C18 chromatographic column (2.1 mm × 50 mm, 2 μm particle size, Advanced Materials Technology, Wilmington, DE, USA), with a flow rate of 0.3 mL/min, and column temperature of 35 °C. The mobile phase consisted of 5 mmol/L ammonium formate containing 0.4% formic acid as phase A and methanol as phase B, and the gradient program was as follows: 0–1.0 min, 25% B; 1.0–3.0 min, 25–95% B; 3.0–5.0 min, 95% B; 5–6 min, 100–25% B; and 6–7.5 min 25% B. The total running time was 9 min. The mass spectrometry parameters were as follows: ion source: ion funnel, electrospray ionization source, and positive ion mode; scanning mode and multiple reaction monitoring, the MRM conditions for the target compounds are given in Table 6; gas temperature, 200 °C; gas flow, 15 L/min; nebulizer, 40 psi; sheath gas temperature, 350 °C; sheath gas flow, 12 L/min; and capillary voltage, 35 psi.

### 3.6. Method Validation

The analytical method used for the analysis of fluopyram and fluopyram-benzamide in carrot roots, leaves, and soils was evaluated by assessing the validation parameters, i.e., recovery, limit of detection (LOD), limit of quantification (LOQ), linearity, and precision.

The performance criteria developed for fluopyram and fluopyram benzamide were validated against the literature [13]. Blank samples of each matrix served as controls. The linearity of the standard calibration solutions prepared at concentrations of 0.005 mg/kg, 0.01 mg/kg, 0.05 mg/kg, 0.10 mg/kg, 0.50 mg/kg, and 1.00 mg/kg was investigated, and the correlation coefficient (R^2^) was calculated. The LODs and LOQs were determined based on the signal-to-noise (S/N) method. The LOD and LOQ were calculated through an S/N of 3 and s S/N of 10 in the spiked recovery experiment, respectively [27,28].

### 3.7. Calculation of Half-Life

The degradation of fluopyram in soil, carrot root, and carrot leaves over time was evaluated by a first-order kinetic equation [29]. It was calculated by the following formula: Ct = C0e ^− kt^ and t_1/2_ = ln 2/k, in which Ct is fluopyram residue concentration at time t, C0 is the initial concentration of fluopyram (mg/kg), k denotes the rate constant of degradation, and t_1/2_ represents the half-life of fluopyram [30].

### 3.8. Dietary Risk Assessment

The dietary risk assessment combined the national estimated daily intake (NEDI) with the risk quotient (RQ). NEDI and RQ are calculated by the following equations [31]:NEDI = ∑ (STMRi × Fi)/bw(1)
RQ% = NEDI/ADI × 100%(2)
where STMRi (mg/kg) represents the supervised trials’ median residue values of fluopyram in carrots from China. If a suitable STMR was not available, the corresponding MRL could be used instead; Fi refers to the food consumption data of a certain agricultural product or food in China [32]; bw (kg) is the average body weight for a population age group [33], and RQ is the chronic risk quotient determined by comparing the NEDI and ADI values. The higher the RQ value, the higher the pesticide residue; RQ > 100% means that the food being evaluated has an unacceptably high health risk to consumers, and the ADI is the acceptable daily intake (0.01 mg/kg bw) [34].

## 4. Conclusions

The residue levels of fluopyram in carrot and soil following root irrigation and spray application were analyzed by LC-QQQ-MS/MS, and fluopyram-benzamide, the main metabolite of fluopyram, was not detected during the entire field trial. In the root irrigation field trial, the initial deposition of fluopyram declined in the order of soil > carrot leaves > carrot roots for treatments at 250.00 g a.i./ha and 500.00 g a.i./ha; in the spray field trail, the initial deposition of fluopyram decreased in the order of carrot leaves > soil > carrot roots. Among the two application methods, the final residues of fluopyram in the edible part of carrot (carrot root) after root irrigation were greater than those after spray application. The final residues of fluopyram in carrot roots were 0.60 mg/kg and 1.80 mg/kg after root irrigation application, while spray application was only 0.05 mg/kg. Most importantly, the provisional maximum residue limit (MRL) of fluopyram in carrots in China is 0.4 mg/kg, and it was obvious that the final residues of fluopyram in carrot roots were much higher than the MRL after root irrigation. However, the RQ values showed that the dietary risk of fluopyram in carrots for consumers from the two applications were at acceptable levels (RQ < 100%). This study revealed the degradation and residual distribution of fluopyram in carrots. These data will guide the correct and safe use of fluopyram for carrot crops. However, the metabolic mechanism of fluopyram in carrots is still unclear, and the transfer of fluopyram in carrots needs to be further studied.

## Figures and Tables

**Figure 1 molecules-27-05544-f001:**
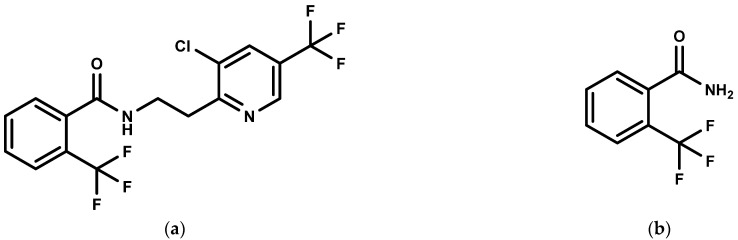
(**a**) The chemical structure of fluopyram; (**b**) The chemical structure of fluopyram-benzamide.

**Figure 2 molecules-27-05544-f002:**
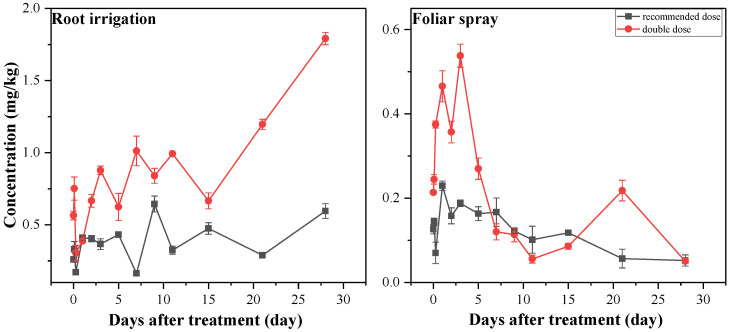
Fluopyram residue concentration in carrot roots expressed as the average of three biological replicates ± standard deviation.

**Table 1 molecules-27-05544-t001:** Recovery and precision of fluopyram and fluopyram benzamide from carrots and soils measured by LC-QQQ-MS/MS.

Sample	Spiking Concentration (mg/kg)	Percent Recovery ± SD (%)	RSD (%)
Fluopyram			
carrot root	0.01	106 ± 5	5
	0.5	83 ± 9	10
	1	102 ± 3	3
carrot leaf	0.01	105 ± 16	15
	0.5	84 ± 2	3
	1	83 ± 4	5
soil	0.01	96 ± 2	3
	0.5	98 ± 10	10
	1	102 ± 4	4
Fluopyram-benzamide			
carrot root	0.01	116 ± 3	3
	0.5	95.9 ± 0.4	0.4
	1	100 ± 4	4
carrot leaf	0.01	115 ± 4	4
	0.5	89 ± 4	4
	1	89 ± 4	5
soil	0.01	80 ± 1	1
	0.5	93 ± 8	9
	1	98 ± 4	4

**Table 2 molecules-27-05544-t002:** Residues of fluopyram in the edible part (carrot root).

Days after Treatment		Residues ± SD (mg/kg)				
		Untreated Control	Root Irrigation		Foliar Spray	
			Treatment at 250.00 g a.i./ha	Treatment at 500.00 g a.i./ha	Treatment at 250.00 g a.i./ha	Treatment at 500.00 g a.i./ha
0	(10 min)	ND	0.26 ± 0.01	0.56 ± 0.03	0.13 ± 0.01	0.213 ± 0.003
	(2 h)	ND	0.33 ± 0.05	0.75 ± 0.08	0.14 ± 0.01	0.24 ± 0.01
	(6 h)	ND	0.170 ± 0.004	0.30 ± 0.05	0.07 ± 0.03	0.38 ± 0.01
1		ND	0.41 ± 0.02	0.39 ± 0.01	0.23 ± 0.01	0.47 ± 0.04
2		ND	0.40 ± 0.02	0.67 ± 0.05	0.16 ± 0.02	0.36 ± 0.03
3		ND	0.37 ± 0.04	0.88 ± 0.03	0.19 ± 0.01	0.54 ± 0.03
5		ND	0.43 ± 0.01	0.62 ± 0.09	0.16 ± 0.02	0.27 ± 0.03
7		ND	0.16 ± 0.02	1.01 ± 0.10	0.17 ± 0.03	0.12 ± 0.02
9		ND	0.64 ± 0.06	0.84 ± 0.05	0.12 ± 0.01	0.11 ± 0.02
11		ND	0.32 ± 0.03	0.99 ± 0.01	0.10 ± 0.03	0.06 ± 0.01
15		ND	0.47 ± 0.04	0.67 ± 0.06	0.118 ± 0.002	0.09 ± 0.01
21		ND	0.30 ± 0.02	1.20 ± 0.04	0.06 ± 0.02	0.22 ± 0.02
28		ND	0.60 ± 0.05	1.80 ± 0.04	0.05 ± 0.01	0.05 ± 0.01

ND = not detected.

**Table 3 molecules-27-05544-t003:** Residues of fluopyram in the carrot leaf.

Days after Treatment		Residues ± SD (mg/kg)				
		Untreated Control	Root Irrigation		Foliar Spray	
			Treatment at 250.00 g a.i./ha	Treatment at 500.00 g a.i./ha	Treatment at 250.00 g a.i./ha	Treatment at 500.00 g a.i./ha
0	(10 min)	ND	1.64 ± 0.05	3.37 ± 0.42	12.62 ± 0.78	23.55 ± 0.08
	(2 h)	ND	1.31 ± 0.11	2.80 ± 0.06	13.52 ± 0.22	25.29 ± 0.18
	(6 h)	ND	0.58 ± 0.03	1.25 ± 0.06	14.15 ± 0.18	27.48 ± 0.28
1		ND	0.51 ± 0.01	1.26 ± 0.14	11.45 ± 0.35	17.77 ± 0.14
2		ND	0.64 ± 0.22	1.28 ± 0.07	8.63 ± 0.27	17.53 ± 0.51
3		ND	1.45 ± 0.12	2.37 ± 0.18	9.32 ± 0.55	15.91 ± 0.55
5		ND	0.41 ± 0.01	1.03 ± 0.03	5.82 ± 0.26	11.72 ± 0.27
7		ND	0.31 ± 0.02	0.77 ± 0.08	1.37 ± 0.04	1.70 ± 0.03
9		ND	0.37 ± 0.02	0.91 ± 0.06	1.42 ± 0.08	1.76 ± 0.02
11		ND	0.49 ± 0.01	0.68 ± 0.03	1.55 ± 0.04	1.44 ± 0.11
15		ND	0.27 ± 0.01	0.73 ± 0.01	0.97 ± 0.21	1.26 ± 0.01
21		ND	0.16 ± 0.01	0.50 ± 0.01	0.12 ± 0.04	1.47 ± 0.06
28		ND	0.13 ± 0.01	0.53 ± 0.01	1.10 ± 0.03	1.44 ± 0.03

ND = not detected.

**Table 4 molecules-27-05544-t004:** Dynamic models of fluopyram in carrot samples from the two application methods.

Application Method	Treatment Concentration	Matrix	Dynamic Equation	Half-Life (d)
Root irrigation	250.00 g a.i./ha	carrot root	-	-
		carrot leaf	Ct = 0.8736 e^−0.076t^	9.1
	500.00 g a.i./ha	carrot root	-	-
		carrot leaf	Ct = 1.6557 e^−0.048t^	14.4
Foliar spray	250.00 g a.i./ha	carrot root	Ct = 0.1963 e^−0.049t^	14.1
		carrot leaf	Ct = 9.108 e^−0.108t^	6.4
	500.00 g a.i./ha	carrot root	Ct = 0.3059 e^−0.063t^	11
		carrot leaf	Ct = 15.868 e^−0.125t^	5.5

**Table 5 molecules-27-05544-t005:** The dietary risk assessment of fluopyram in carrot samples.

Age (Years)	Sex	Body Weight (kg)	Fi(g/d)	NEDI (mg/kg bw)		RQ (%)	
				Root Irrigation	Foliar Spray	Root Irrigation	Foliar Spray
2–7		17.9	194.8	0.07	0.025	40	14
8–12		33.1	272.4	0.10	0.034	30	10
13–19	M	56.4	396.7	0.15	0.050	26	9
	F	50	317.9	0.12	0.040	23	8
20–50	M	63	436.4	0.16	0.055	25	9
	F	56	412.1	0.15	0.052	27	9
51–65	M	65	477.9	0.18	0.060	27	9
	F	58	447	0.16	0.056	28	10
>65	M	59.5	413.3	0.15	0.052	25	9
	F	52	364.1	0.13	0.046	26	9

M = male; F = female; Fi = the food consumption data of a certain agricultural product or food in China; NEDI = national estimated daily intake; RQ = risk quotient.

**Table 6 molecules-27-05544-t006:** Multiple reaction monitoring (MRM) conditions.

Name	Molecular Formula	Retention Time (min)	Precursor Ion (*m*/*z*)	Product Ion (*m*/*z*)	Collision Energy (V)
Fluopyram	C_16_H_11_ClF_6_N_2_O	3.3	397	207	25
				173	25
Fluopyram- benzamide	C_8_H_6_ClF_3_NO	1.3	190	170	10
				130	25

## Data Availability

The data presented in this study are available on request from the corresponding author.

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
