# Peer review of "Residue and Risk Assessment of Fluopyram in Carrot Tissues"

_molecules, 2022, doi:10.3390/molecules27175544_

Round 1
Reviewer 1 Report
The manuscript deals with actual problem on the example of the determination of fluopyram residual concentrations and its principal metabolite (fluopyram-benzamide) in carrot roots. The principal movelty is comparing these concentrations in the results of two different application methods, namely root irrigation and foliar spray. The manuscript is well presented and it contains all discussion points required for papers of such design.
However, the reviewer’s principal disagreement with the text presented is the number of significant digits in numerous values. Let us consider this problem step-by-step through the whole text.
At first, the LOD and LOQ values mentioned in the Abstract are 0.047-2.65 and 0.156-8.818 ug/kg. The indication of any values with FOUR significant digits seems to be inacceptable. The first reason for that is the number of such digits in the amounts of fluopyram used for carrot processing: 250 or 500 g a.i./ha (only two significant digits). The number of significant digits in the results cannot exceed their number in initial data. The second reason is the following: if the analytical result is presented with FOUR significant digits (8.818), it means that the analytical method used provides the relative precision of the results approx. 0.01%. The reviewer is not sure that LC-QQQ-MS/MS quantification corresponds to this condition.
Abstract again: final fluopyram residues were 0.595 and 1.791 mg/g. The same comments: too many significant digits. Risk quotient is 23.3% - 39.8%. Why not 23 – 40% ? And the next values 8.01% - 13.71%. Why not 8 – 14% ? The larger number of digits looks more scientific?
Page 2, section 2.1: 82.70% - 105.76%, 79.57% - 116.33%, and 0.156-8.818 again. Look that the values 105.76 and 116.33 contain FIVE significant digits.
Table 1 (pages 2-3) looks like terrible paradox. The spiking concentrations (column 2) are 0.01, 0.5, and 1 mg/kg, i.e. these values are indicated with only ONE significant digit. At the same time, the percent recoveries +- SD (%) (column 3) are indicated with 4-5 significant digits, and RSD (%) (column 4) with 3-4 such digits. Moreover, the number format, e.g., 105.76 +- 5.18% is absolutely inappropriate. If the uncertainty of this number exceeds 5%, the indication of decimal (or hundreds) parts is reasonless. It should be 106 +- 5. Hence, all the data in the columns 3 – 5 of this Table should be rounded up to not more than two significant digits.
Page 3, section 2.2: the values 5.342, 14.461, 0.176, 0.332, 6.018, 13.832, 0.295, and 0.593 should be rounded, as well.
Table 3 (pages 3-4): most of data in columns 3, 4, 5, and 6 should be rounded. Please do that carefully, because, for example, the value 0.332 +- 0.006 is absolutely correct.
Page 4, section 2.3: the values 0.259, 0.126, 0.563, 0.213, 0.595, 1.792 should be rounded.
Table 3 (page 4): most of data (but not all of them) should be rounded.
Page 5, section 2.4: the similar rounding is required.
Table 4 (page 5): most of data (but not all of them) should be rounded.
Page 5, section 2.5: the half-lives of fluopyram in pomegranate fruit are 7.6 and 8.6 days, while in watermelon it was 6.48 days. Why half-life for watermelon was determined with better precision that that for pomegranate? It looks at least funny.
Page 5, section 2.6: the values 8.01 and 39.83 should be rounded.
Table 6 (page 6-7): the data in the column “NEDI; Foliar spray” are out of criticism. They consist of two significant digits. All other data should be rounded.
The additional remarks are the following:
The footnote to Table 6 include explanation “M = Male; F = Female”. Of course, this commonly known symbolism should be explained. However, in more extent the decoding is required for the symbols FI(g/d) and NEDI in the columns names of the same table. Of course, the sense of NEDI is explained at page 8, but the first time this abbreviation is appeared at page 6.
Additionally the “commonly unknown” symbolism “a.i./ha” (first time it is appeared in the Abstract) should be explained nearly at its first mention.
Thus, the principal problem of this manuscript is the unreliably large number of significant digits in the most of numerical data. All of them should be rounded up to appropriate amount of such digits. The reviewer cannot imagine the publication of this paper without rounding.
Author Response
Dear editor,
Thank you for editor’ and reviewers’ opinions, these comments are very helpful to improve the quality of the manuscript. We have carefully revised our manuscript for improving the quality of the manuscript. Words in blue are the changes I have made in the manuscript. Now I response the reviewer’ comments with a point by point and highlight the changes in revised manuscript. Full details of the files are listed. We sincerely hope that you find our responses and modifications satisfactory.
Comment #1: The principal problem of this manuscript is the unreliably large number of significant digits in the most of numerical data. All of them should be rounded up to appropriate amount of such digits.
Response: Thanks for reviewer’s comments, we are very sorry for our careless mistake in numerical data, and it was rectified in manuscript.
Comment #2: Page 5, section 2.5: the half-lives of fluopyram in pomegranate fruit are 7.6 and 8.6 days, while in watermelon it was 6.48 days. Why half-life for watermelon was determined with better precision that that for pomegranate?
Response: Regarding the half-life of fluopyram in different crops, it may be due to the difference in the number of significant digits retained by the authors when processing the data, we directly quoted the data given in the reference.
Comment #3: However, in more extent the decoding is required for the symbols FI(g/d) and NEDI in the columns names of the same table. Of course, the sense of NEDI is explained at page 8, but the first time this abbreviation is appeared at page 6.
响应: 我们非常感谢这个好的建议,我们按照您的想法做了:“Fi=中国某种农产品或食品的食品消费数据;NEDI=全国估计每日摄入量;RQ= 风险商数。在第176-177行。
评论#4:此外,“通常未知的”象征意义“a.i./ha”(第一次出现在摘要中)应该在第一次提到时解释。
响应:感谢审稿人的评论,我们在手稿的摘要部分修改为:第12行的“每公顷活性成分(a.i./ha)”。
Reviewer 2 Report
I would like to see more analytical method information. For example QuEChERS method employed in this work. Sample preparation information the authors provided not sufficient.

Author Response
Dear editor,
Thank you for editor’ and reviewers’ opinions, these comments are very helpful to improve the quality of the manuscript. We have carefully revised our manuscript for improving the quality of the manuscript. Words in blue are the changes I have made in the manuscript. Now I response the reviewer’ comments with a point by point and highlight the changes in revised manuscript. Full details of the files are listed. We sincerely hope that you find our responses and modifications satisfactory.
Comment #1:
Organizing the manuscript: In this manuscript the first section is the “Introduction” and the second is the “Results and Discussion”. I would like to see a separate section on “Materials and Methods” between the “Introduction” section and “Results and Discussion” sections. I believe the authors did not provide sufficient experimental details for the reader. The section “Materials and Methods” should be written is a way that another scientist can reproduce this work.
Response: Thanks for reviewer’s comments, and the format of this manuscript is modified according to the template given by the journal. And the experimental details was given in line 178-261.
Comment #2:
QuEChERS method: The authors employed a QuEChERS sample preparation method for this work. There are many variations of the method (regarding the salting out step and solvent extraction involved). Please provide details.
Response: We appreciate it very much for this good suggestion, the QuEChERS sample preparation method different from the previous, because we referred the literature [13,19] in manuscript line 211. And finally, all recoveries were within the acceptable range of 80-146 %, so we were not optimizing the method.
Comment #3:
Introduction section:
I believe the authors have not given sufficient background information previous research work. Several authors have reported several methods in the past for various other plant types [1]. Please provide sufficient background information previous similar methods and what explain what is unique about this work.
Response: Thanks for reviewer’s comments, the reference given by reviewer is to use the HRMS examined the metabolic degradation of five pesticides after hydroponic root uptake situation, but this research is to use LC-QQQ-MS/MS examined the residue of fluopyram and its metabolite. We think the research direction between the reference and this research is different.

Reviewer 3 Report
Manuscript title: Residue and risk assessment of fluopyram in carrot tissues
This study evaluated the variation in residue behavior of fluopyram in soil, carrot root and carrot leaf samples after application of fluopyram.It seems to me that the study is very important and original.
Author Response
感谢您的评论!